# Predictors in Optic Pathway Gliomas in Neurofibromatosis Type 1: A Single Center Study

**DOI:** 10.3390/cancers17091404

**Published:** 2025-04-23

**Authors:** Agata Marjańska, Jagoda Styczyńska, Agnieszka Jatczak-Gaca, Joanna Stachura, Michał Marjański, Jan Styczyński

**Affiliations:** 1Department of Pediatric Hematology and Oncology, Collegium Medicum, Nicolaus Copernicus University Torun, Jurasz University Hospital, 85-095 Bydgoszcz, Poland; jagasadlok@gmail.com (J.S.); agaca05@tlen.pl (A.J.-G.); jstyczynski@cm.umk.pl (J.S.); 2Department of Ophthalmology, Collegium Medicum, Nicolaus Copernicus University Torun, Jurasz University Hospital, 85-095 Bydgoszcz, Poland; joanna.stachura@cm.umk.pl; 3Faculty of Medicine, Department of Clinical Oncology, Oncology Center, Bydgoszcz University of Science and Technology, 85-796 Bydgoszcz, Poland; michalmarjanski@tlen.pl

**Keywords:** neurofibromatosis type 1, NF1, children, optic pathway glioma, chemotherapy

## Abstract

Optic pathway gliomas (OPG) are typical tumors of young patients with neurofibromatosis type 1 (NF1). The principles of diagnosis and treatment have remained unchanged for decades. We analyzed clinical data on 92 patients (demographics, tumor anatomy, ophthalmological and imaging examination results, applied therapies and their effects). Risk factor analyses for amblyopia and necessity of oncological treatment were performed. Based on the analyses, the authors suggest modification of the current guidelines for the management of patients with NF1-OPG, including the abandonment of contrast administration in control magnetic resonance imaging (MRI), reducing the frequency of MRI in patients with isolated involvement of the initial nerve segment by the tumor, and a cautious approach to patients with comorbidities such as delayed psychomotor development or epilepsy.

## 1. Introduction

Neurofibromatosis type 1 (NF1) is a genetically determined disorder occurring with a frequency of 1:2500. An increased risk of neoplasia is typical for the disease, and among NF1-dependent tumors, one of the most common are optic pathway gliomas (OPGs) [1].

OPGs are tumors of the central nervous system (CNS) located in any part of the optic pathway (Figure 1) [2,3]. In the histopathological evaluation, they are usually described as pilocytic astrocytoma grade I according to the WHO classification [4].

It is estimated that OPGs affect 15–20% of patients with NF1. They usually develop between 2 and 8 years of age. Some studies suggest a more frequent occurrence of OPGs in females. An increased risk of OPG has been demonstrated in patients with selected mutations in the Nf1 gene (p.Gly848, p.Leu874, p.Ala846, p.Cys845, p.Leu844, p.Met992) [4,5,6]. Some reports suggest that in patients with NF1 and co-existing history of allergic disease, the risk of developing OPG may be reduced [7,8]. Despite the complexity of the course of NF1 and the co-occurrence of other diseases in individuals with OPG, there is a lack of data regarding their mutual influence and predisposition to common occurrence.

In most cases, OPGs are asymptomatic for a long time. Due to their slow growth tendency, they are rarely mortal. The most important symptom resulting from their presence is the risk of partial or complete loss of vision. Due to the location of the visual pathway close to the hypothalamus and pituitary gland, OPGs can lead to endocrine disorders [4,9,10].

Neuroimaging using magnetic resonance (MRI) of the orbits is recommended for the diagnosis of OPGs. The diagnosis is made when fusiform dilatation of any part of the optic nerve, sometimes with concomitant tortuous course, or thickening of the chiasm or tracts is observed. OPGs on MRI have high signal on T2-weighted images, but the tumor contrast is variable. However, the decisive parameter for the implementation of oncological treatment is deterioration in a full ophthalmological examination assessing visual acuity (VA), field of vision, Visual Evoked Potentials (VEPs) and Optical Coherence Tomography (OCT). Age-appropriate VA assessment is crucial for the surveillance of NF1-OPG. Teller acuity cards, eye chart with standard letters (HOTV) or shapes (Lea), and Snellen acuity cards can be used for patients aged 0 to 2 years, 2 to 6 years, and 6 to 15 years, respectively. According to current guidelines, VA assessment plays a major role in determining disease progression and treatment decisions. Therefore, serial VA measurements are recommended for the follow-up of affected patients [1,4,11].

The standard strategy for NF1-associated OPGs is the “watch and wait” approach. Indications for treatment resulting from significant clinical or radiological progression are observed in one third of patients [4]. The behavior of NF1-OPGs can be unpredictable and there are no clear prognostic features. Some factors have been presented as associated with an increased risk of clinical progression: female gender, age < 2 years and >8 years, and tumor location in the post-chiasmatic optic pathway [9]. The decision to start treatment is multifactorial and should be the result of evaluation by a multidisciplinary team consisting of an oncologist, an ophthalmologist, and a radiologist. In the first line of treatment, chemotherapy based on vincristine and carboplatin is preferred. In subsequent lines, vinblastine, antiangiogenic drugs (bevacizumab), and irinotecan are used. Recently, promising effects of targeted therapies using MEK inhibitors (selumetinib, trametinib) have been observed in ongoing clinical trials [12,13,14,15,16].

The estimated 5-year OS for patients with OPG is 96–100%. Deaths resulting from tumor progression are incidental. Premature deaths are mainly related to late complications after treatment, especially from secondary tumors after radiotherapy [17].

The objective of this study was the analysis of the course, risk factors, indications for treatment, and effects of therapy for OPGs in NF1 patients.

## 2. Materials and Methods

*Study design.* The retrospective single-center analysis was performed on large cohort of NF1-patients with diagnosed OPG, with full clinical data available.

*Patients.* The Neurofibromatosis Center at the University Hospital No. 1 in Bydgoszcz is the largest center coordinating care for patients with NF1 in Poland. Between 1999–2024, the Center provided care to 1048 patients with NF1. All NF1-individuals with OPGs (*n* = 164) were selected from the entire group. The complete medical history of each person from this cohort was carefully reviewed, ultimately including 92 patients with full documentation including the results of subsequent MRI examinations and ophthalmological evaluations (Figure 2). Inclusion criteria were: (1) NF1 diagnosis based on the NIH criteria from 1988 [18] or updated diagnostic criteria from 2021 [19], (2) OPG diagnosis based on MRI. Exclusion criteria were: (1) lack of NF1 diagnosis, (2) lack of OPG diagnosis, (3) lack of complete descriptions of MRI results, (4) lack of access to complete descriptions of ophthalmological consultations.

*Analyzed factors.* Primary endpoint was necessity of oncological treatment of OPG. Secondary endpoint was development of amblyopia. We analyzed patient demographics, imaging and ophthalmological examination results, and their impact on further therapeutic decisions and the effectiveness of used therapy. Additionally, all available clinical and genetic data were assessed. The following variables were analyzed: sex (male vs. female); age (<3.5 vs. ≥3.5 years); OPG localization (nerve involvement divided into individual segments, chiasm or tract); OPG features in MRI (post-contrast enhancement, maximum nerve thickness with the nerve sheath <8 mm vs. ≥8 mm, unilateral vs. bilateral OPG); ophthalmological examination results including amblyopia, strabismus, and exophthalmos; presence of Lisch nodules; form of NF1 (familial vs. de novo); presence of comorbidities such as: delayed psychomotor development, epilepsy, allergy, presence of plexiform neurofibroma of any location, heart defects, hydrocephalus, endocrine dysfunctions with a possible relationship with OPG (precocious puberty, short stature), hypertension, autism spectrum disorders, attention deficit hyperactivity disorder (ADHD), presence of pseudoarthrosis. The study was approved by the Institutional Review Board.

*Statistical analysis.* Data were expressed as medians and ranges. Baseline parameters were compared using Chi-square or Fisher’s exact tests for quantitative variables and Mann–Whitney test for continuous variables. Univariate and multivariate risk factor analyses were performed using logistic regression model with odds ratio (OR) and 95% CI (confidence intervals). Factors with *p*-value < 0.1 in univariate analysis were analyzed in multivariate analysis. Data were computed using SPSS 29 (IBM SPSS Statistics, Version 29.0). Statistical significance was regarded as *p* < 0.05.

## 3. Results

### 3.1. Demographics and Clinical Profile

A total of 92 patients were retrospectively analyzed, including 57% female (52/92) and 43% male (40/92). The median age at OPG diagnosis was 3.4 years (range: 1.0–24.1 years). In 38% (35/92) of patients, NF1 was familial, and de novo in 62% (57/92). In 21 patients the disorder was genetically confirmed, and mutations found in the Nf1 gene are presented in Table 1. In 16.3% of patients, OPG was diagnosed before 2 years of age. Comorbidities are presented in Table 2.

### 3.2. Characteristics of Optic Pathway Gliomas

Isolated involvement of the optic nerve by the tumor was observed in 64.4% (62/92) of patients. OPG in the nerve together with the chiasm was described in 14.1% (13/92) of individuals. Infiltration of the chiasm and optic tract (without nerve) and isolated tract involvement was observed in 3.3% (3/92) and 2.2% (2/92) patients, respectively. Tumor growth involving the nerve, chiasm, and tract was observed in 13% (12/92) of patients. In addition, optic nerve segment involvement was assessed and the following results were obtained: intraocular segment in 5.4% (5/92), intraorbital segment in 64.1% (59/92), intracanalicular segment in 46.7% (43/92), intracranial segment in 58.7% (54/92). Chiasmal invasion (irrespective of involvement of other optic pathway segments) was described in a total of 30.4% (28/92) of patients, and optic tract in 18.5% (17/92). In 55.4% (51/92) of patients, OPGs were unilateral (right-sided in 20/51 and left-sided in 31/51), while bilateral in 44.6% (41/92). Post-contrast enhancement in MRI was observed in 67.4% (62/92). Abnormalities in the ophthalmological examination included: strabismus in 28.3% (26/92), proptosis in 9.8% (9/92), partial amblyopia in 33.7% (31/92), and complete amblyopia in 3.3% (3/92). Co-occurrence of Lisch nodules with OPGs was present in 42.4% (39/92), and among patients undergoing oncological treatment, Lisch nodules were visible in 8/15 individuals.

Among patients diagnosed with OPG under the age of 2 years, an isolated involvement of the optic nerve by the tumor was observed in 75% (12/16) of patients, OPG in the nerve together with the chiasm was described in 18.8% (3/16) of individuals, and the tumor growth involving the nerve, chiasm, and tract was observed in 6.3% (1/16) of patients. OPGs were bilateral in 50% (8/16) of patients. Post-contrast enhancement in MRI was observed in 93.8% (15/16). Abnormalities in the ophthalmological examination included: strabismus with proptosis in 12.5% (2/16), partial amblyopia in 31.3% (5/16), and complete amblyopia in 6.3% (1/16).

### 3.3. Indications for Oncological Treatment

Oncological treatment due to progression of OPG was required in 16.3% (15/92) of patients, including eleven females and four males. The median age of individuals at the time of diagnosis OPG was 2.9 years (range: 1.0–7.1 years). For comparison, the median age of patients at OPG diagnosis who did not undergo oncological therapy was 3.6 years (range: 1.3–24.1 years). The median age of patients at the time of treatment initiation was 3.8 years (range: 1.5–10.3 years). The median time from tumor diagnosis to oncological treatment was 6.5 months (range: 0.3–47 months). Due to the sudden occurrence of proptosis in 3/15 patients, treatment was initiated promptly. In 5/15 patients, the decision of therapy was made after more than one year, and in 1/5 of them after 4 years (due to very slowly but gradually progressing vision loss). The reasons for treatment decisions were as follows: deterioration of ophthalmological parameters with accompanying progression of size in MRI concerned 7/15 patients, isolated deterioration of vision (without progression in MRI) in 4/15 patients, and isolated progression in MRI (without obvious deterioration of vision) in 4/15 patients. Among symptomatic patients: 10/26 patients with strabismus, 6/9 patients with proptosis, and 3/3 patients with complete amblyopia required oncological treatment.

Univariate analysis indicated the influence of nine factors on the necessity of oncological therapy in patients with OPG including: proptosis, strabismus, amblyopia, co-occurrence of epilepsy, thickness of the optic nerve ≥ 8 mm, bilateral involvement of visual pathway by OPG, involvement of the following segments of the visual pathway by the tumor: intracranial and/or intracanalicular segment of nerve, or chiasm. (Table 3). In multivariate analysis two factors were statistically significant: amblyopia (OR = 9.5; 95% CI = 2.19–41.7; *p* < 0.01) and proptosis (OR = 13.9; 95% CI = 2.37–82.1; *p* < 0.01) (Table 4). No interdependence was found between the occurrence of post-contrast enhancement in the MRI and the need for oncological treatment. Contribution between the involvement of intraocular and intraorbital fragments of optic nerve and the need for oncological therapy was also not found in the analyzed study group.

In univariate analysis, the following factors had significant effect on amblyopia in patients with NF1-OPG: strabismus, co-occurrence of epilepsy or delayed psychomotor development, thickness of the optic nerve ≥ 8 mm, bilateral involvement of visual pathway by OPG, involvement of the following segments of the visual pathway by the tumor: intracranial and/or intracanalicular segment of nerve, chiasm, and familial type of NF1 (Table 5). Factors contributing to amblyopia in multivariate analysis include: bilateral involvement of the visual pathway by OPG, thickness of the optic nerve ≥ 8 mm, strabismus, proptosis, and the occurrence of epilepsy (Table 6).

In all patients undergoing oncological treatment, OPG involved the optic nerve—isolated in 7/15, with concomitant involvement of the chiasm +/− tract in 8/15 cases. In all individuals with isolated optic nerve involvement, the infiltration involved the intraorbital fragment of the nerve. The median thickness of the optic nerve for 15 patients with oncological therapy was 9.2 mm (range: 5–14.5 mm). In 11/15 patients, OPG was bilateral. OPG did not show post-contrast enhancement on MRI in 2/15 patients requiring therapy.

### 3.4. Oncological Therapy

The first line of oncological treatment of OPG included: vincristine + carboplatin (VCR + Carbo; treatment time: 1 year) regimen in 8/15, monotherapy with vinblastine (VBL; treatment time: 60 weeks) in 6/15 and radiotherapy in 1/15 patients (this patient was treated abroad and the reason for choosing this method is unclear). Almost half of the patients (7/15) required subsequent lines of oncological treatment: three lines in 3/15, four lines in 2/15, and five lines in 1/15. Subsequent lines after using VCR + Carbo and VBL included: a combination of bevacizumab with irinotecan, temozolamide in monotherapy, or selumetinib. Among patients treated with VBL, 36.4% (4/11) required subsequent lines of therapy, and among patients treated with VCR + Carbo, subsequent lines of therapy were required in 66.7% (6/9) of them. In all patients with a good response to first-line therapy, partial regression of tumors was observed in MRI with a concomitant partial improvement of the ophthalmological examination result. None of the patients without a response to first-line therapy achieved clinical improvement after the use of subsequent lines of therapy. The median age of patients at diagnosis with OPG response to first-line therapy and without response to first-line therapy was 2.8 vs. 2.9 years (*p* = ns), respectively. In the group of patients treated with VBL, 36.4% (4/11) required dose reduction due to persistent neutropenia. In the group of patients treated with the VCR + Carbo regimen, severe complications (i.e., III and IV CTCAE grade) were observed in 55.6% (5/9) patients and included anaphylaxis (*n* = 3), severe polyneuropathy (*n* = 1), and acute hemolysis (*n* = 1).

## 4. Discussion

Diagnostics and treatment of OPGs have remained unchanged for years. The available guidelines for the management of patients with NF1 recommend performing the first imaging examination of the CNS after the age of 2 years, unless ophthalmological assessment is suggestive for OPG [1,10]. The median age of patients starting oncological treatment in our analysis was 3.8 years, but in 16/92 patients OPG was diagnosed before 2 years of age, and in 2/15 oncological treatment was initiated earlier than 2 years of age. Abnormalities in ophthalmological examination were found in 37.5% (6/16) of patients, but in 62.5% (10/16) OPG was an incidental finding. Additionally, a reliable ophthalmological examination is often impossible to perform in the youngest patients. Early detection of OPG can prevent vision loss in the youngest patients. Therefore, it seems reasonable to perform the first MRI after the age of 1 year.

There is no consensus on the frequency of MRI after OPG diagnosis. It depends on age, presence of symptoms, initiation of therapy and the views of the investigators. Follow-up definitely includes clinical evaluation. Based on selected recommendations, after the diagnosis of OPG (regardless of location), subsequent MRI should be performed every 3 months in the first year of observation [1,10]. In our study, we precisely analyzed the relationship between the location of OPGs, the occurrence of amblyopia and the need for oncological therapy. Both univariate and multivariate analyses showed low clinical aggressiveness of tumors located in the initial segment of the optic nerve. In 25/92 patients there was an isolated involvement of the nerve in the intraocular and/or intraorbital segments, and only in 1/25 partial amblyopia was found (patient after radiotherapy abroad). It seems reasonable to abandon such a high frequency of follow-up examinations in patients with the presence of OPGs in the initial segments of the optic nerves.

Despite the high safety of MRI examinations compared to computed tomography resulting from the elimination of harmful X-rays, there are emerging reports of an increased risk of dementia in patients undergoing multiple MRI examinations using gadolinium contrast [20,21]. Several previous analyses reported that the administration of contrast material for the assessment of OPG was inappropriate [22,23,24]. Many analyses have confirmed (including this study) that an ophthalmological examination is the clue to making therapeutic decisions [10,11,25,26]. MRI is performed to estimate the size of the tumor, which can be assessed without the use of a contrast agent. Based on our analysis, it is concluded that contrast enhancement in MRI in patients with OPGs has no impact on therapeutic decisions.

The crucial issue in management of OPG is the need of oncological treatment. In multivariate analysis we identified two factors significantly contributing to decision of chemotherapy: amblyopia and proptosis. Amblyopia is the most serious complication of OPG; it leads to severe disability of patients. Implementation of chemotherapy allows to improve vision only in a small percentage of patients. The aim of treatment is to stop vision loss as early as possible. Strabismus in a patient with NF1 is always worrying, because it is associated with large tumors and rapid progression [4,9]. In our study group, 3/9 patients with strabismus did not undergo oncological treatment due to the fact that they came to our center a long time after the occurrence of this symptom, the stabilization of this condition was maintained, and the expected benefits from the implementation of chemotherapy were negligible.

We additionally analyzed risk factors contributing to development of amblyopia. Five factors were related to this complication: bilateral involvement of the visual pathway by OPG, thickness of the optic nerve ≥ 8 mm, strabismus, proptosis, and the occurrence of epilepsy. An important observation in our study is the statistically significant interdependence between the occurrence of epilepsy, amblyopia, and the need for oncological therapy, both in univariate and multivariate analysis. There are reports of the influence of epilepsy and antiepileptic drugs on the deterioration of vision in patients [27,28,29]. Therefore, special caution should be exercised when making decisions about OPG treatment in patients with epilepsy. Similarly, the relationship between delayed psychomotor development and amblyopia. Perhaps the poorer quality of the ophthalmological examination in children with intellectual disabilities gives false positive results in the field of amblyopia, which may ultimately affect a too hasty decision regarding the need for therapy.

Atopic conditions such as comorbidity have been reported to be associated with decreased glioma risk by as much as 20–40% [30]. It has been estimated that allergic diseases in Poland affect 40% of the general population [31]. In the analyzed study group, allergic history was positive in 14.1% of individuals. There are no analyses of the occurrence of allergies in the population of patients with NF1, but the difference between the general population and the population of patients with NF1-OPG is significant. This fact correlates with reports of the protective effect of atopy on the formation of CNS tumors. It is necessary to discover the pathomechanism of this phenomenon, which in the future could be the basis for finding drugs with protective properties in relation to the formation of gliomas in patients with NF1.

Of the patients treated with VBL, 36.4% (4/11) required subsequent lines of therapy. In turn, of the entire group of patients treated with VCR + Carbo, subsequent lines of therapy were required by 66.7% (6/9) of individuals. Answers to what stage should oncological treatment be initiated in patients with NF1-OPG and what therapy should be used in the first and subsequent lines still remain unclear and require further analysis. Lasaletta et al. presented the results of first-line vinblastine treatment in 54 patients with pediatric low-grade gliomas, including 13 NF1-individuals. The 5-year progression-free survival (PFS) in the entire cohort was 53.2%, and for patients with NF1: 85.1% [32]. Kebudi et al. described the results of their own experience with the use of VCR + Carbo in first-line OPG therapy in 95 children (including 63 people with NF1). Ten-year PFS in all patients, in patients with NF1, and without NF1 were 71.6%, 85.7%, and 54.2%, respectively. There are no studies comparing the efficacy of VCR + Carbo vs. VBL in first-line OPG treatment in patients with NF1 [33]. Currently, great hopes are placed in targeted anti-MEK drugs, which, compared to classical chemotherapy, are less toxic and do not require the implantation of a central catheter, which significantly facilitates maintaining a normal lifestyle associated with attending school, physical activity, and the lack of the need for isolation resulting from drug-induced myelosuppression. In our study group, one child is recently treated with selumetinib (currently ninth cycle) as the fifth line of OPG therapy; the treatment is tolerated well, and the size of OPG and the results of ophthalmological tests remain stable.

Our study has some limitations, including being a retrospective and the single-center design of the analysis. It resulted in relatively long-time period of inclusion. Nevertheless, we did not find any differences in clinical course, treatment, and outcome related to time period. Being a single-center study enabled us to follow-up all patients in uniform approach. In order to compare our data, we analyzed reports published in the literature. We identified data on the characteristics of NF1-OPGs presented in published studies (Table 7).

## 5. Conclusions

In summary, based on the collected data, we propose to consider the following modifications in the management of patients with NF1-OPGs: (1) perform the first MRI after the age of 1 year, (2) reduce the frequency of follow-up scans in the first year of observation in patients with isolated involvement of intraocular and/or intraorbital segments of the optic nerve (instead of every 3 months, perform MRI every 4–6 months); (3) refrain from administering contrast during control MRI examinations of the orbits after OPG diagnosis; (4) in patients with co-occurring psychomotor delay or treated with antiepileptic drugs, do not make decisions regarding the implementation of oncological therapy when VA deterioration is observed without progression in OCT, VEP, and MRI examinations.

## Figures and Tables

**Figure 1 cancers-17-01404-f001:**
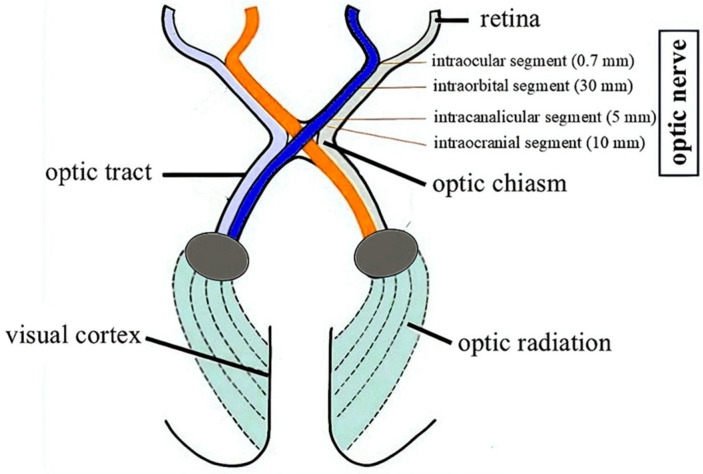
Anatomy of the visual pathway.

**Figure 2 cancers-17-01404-f002:**
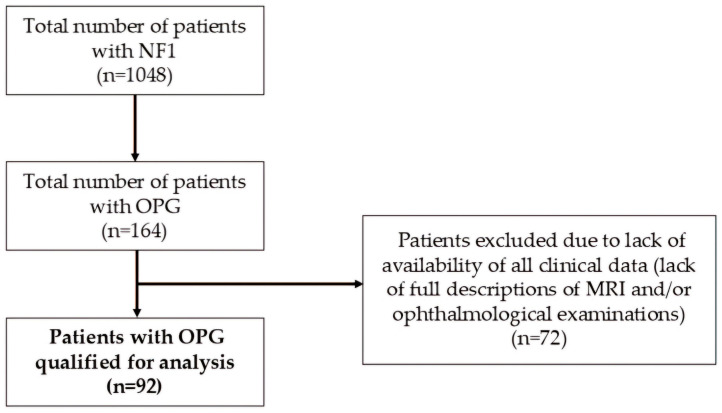
Selection of patients qualified for analysis.

**Table 1 cancers-17-01404-t001:** Mutations in Nf1 gene detected in 21 patients with OPG.

No	Mutation	Optic Pathway Glioma	Additional Information
1	c.4600C>T p.Arg1534*	- localization: optic nerve, unilateral - no oncological treatment - without ophthalmic abnormalities	- allergy - ADHD
2	c.2325G>A p.Glu775=	- localization: optic nerve, unilateral - oncological treatment (1 line) - partial amblyopia	- allergy - autism spectrum disorder - epilepsy - delayed psychomotor development
3	c.4250delG p.Gly1417fs	- localization: optic nerve, bilateral - no oncological treatment - without ophthalmic abnormalities	- delayed psychomotor development
4	c.1392+1G>A/- p.[?]	- localization: optic nerve+chiasm, bilateral - oncological treatment (5 lines) - complete amblyopia, strabismus	- epilepsy - delayed psychomotor development
5	c.1392+1G>A/- p.[?]	- localization: optic nerve, unilateral - no oncological treatment - without ophthalmic abnormalities	- acoustic neuroma
6	c.1392+1G>A/- p.[?]	- localization: optic nerve, bilateral - oncological treatment (2 lines) - partial amblyopia, strabismus	- delayed psychomotor development
7	c.4084C>T p.Arg1362Ter	- localization: chiasm+tract, bilateral - no oncological treatment - without ophthalmic abnormalities	- no additional problems
8	c.1756_1759del p.Thr586Valfs*18	- localization: optic nerve, unilateral - no oncological treatment - without ophthalmic abnormalities	- no additional problems
9	c.5804T>G p.Leu1935Arg	- localization: optic nerve, unilateral - no oncological treatment - without ophthalmic abnormalities	- delayed psychomotor development
10	c.7458-2A>G p.[?]	- localization: optic nerve, bilateral - no oncological treatment - without ophthalmic abnormalities	- no additional problems
11	c.3721C>T p.Arg1241*	- localization: optic nerve + chiasm + tract, bilateral - no oncological treatment - without ophthalmic abnormalities	- no additional problems
12	c.3721C>T p.Arg1241Ter	- localization: optic nerve, bilateral - no oncological treatment - partial amblyopia	- no additional problems
13	c.6349C>T p.Gln2117*	- localization: optic nerve, unilateral - no oncological treatment - without ophthalmic abnormalities	- no additional problems
14	c.1318C>T p.Arg440-*	- localization: optic nerve, unilateral - no oncological treatment - without ophthalmic abnormalities	- no additional problems
15	c.4368-2A>G p.[?]	- localization: optic nerve, unilateral - no oncological treatment - eyelid droop	- no additional problems
16	c.1381C>T p.Arg461Ter	- localization: optic nerve, bilateral - no oncological treatment - without ophthalmic abnormalities	- plexiform neurofibroma of the lower limb
17	c.1061_1062+13delinsGA p.Lys354ARG	- localization: optic nerve, bilateral - no oncological treatment - partial amblyopia, nystagmus	- delayed psychomotor development
18	c.79>T ??? p.Gln27Ter	- localization: optic nerve + chiasm + tract, bilateral - no oncological treatment - partial amblyopia	- delayed psychomotor development
19	c.1541_1542del p.Gln514Argfs*43/-	- localization: optic nerve + chiasm + tract, bilateral - oncological treatment (2 lines) - partial amblyopia, exophthalmos, nystagmus	- delayed psychomotor development
20	c.2146G>T p.Glu716Ter	- localization: optic nerve + chiasm, bilatera - no oncological treatment - partial amblyopia	- Allergy - ASD - ADHD
21	c.3721C>T p.Arg1241*	- localization: optic nerve, unilateral - no oncological treatment - without ophthalmic abnormalities	- no additional problems

ADHD—attention deficit hyperactivity disorder; ASD—atrial septal defect.

**Table 2 cancers-17-01404-t002:** Other comorbidities identified in group of patients with NF1-OPG.

Disorder	Frequency
Delayed psychomotor development	33.7% (31/92)
Allergy	14.1% (13/92)
Plexiform neurofibroma of any location	14.1% (13/92)
Attention deficit hyperactivity disorder (ADHD)	7.6% (7/92)
Epilepsy	7.6% (7/92)
Hydrocephalus	6.5% (6/92)
Congenital heart disease: PS (*n* = 2), PFO (*n* = 1), MR (*n* = 1), ASD (*n* = 1)	5.4% (5/92)
Short stature	4.3% (4/92)
Autism spectrum disorders	3.3% (3/92)
Precocious puberty	3.3% (3/92)
Pseudarthrosis	1.1% (1/92)
Acoustic neuroma	1.1% (1/92)

PS—pulmonary stenosis; PFO—patent foramen ovale; MR—mitral regurgitation; ASD—atrial septal defect.

**Table 3 cancers-17-01404-t003:** Univariate analysis for risk factors of oncological therapy.

Parameter	*p*	OR	95% CI
Thickness of the optic nerve ≥ 8 mm	<0.001	7.63	2.28–25.5
Proptosis	<0.001	16.4	3.50–77.4
Strabismus	<0.001	7.63	2.29–25.5
Amblyopia	<0.001	10.7	2.74–41.6
Epilepsy	0.008	8.97	1.77–45.6
Bilateral involvement of the visual pathway by OPG	0.020	4.31	1.26–14.8
Involvement of intracranial segment by OPG	0.028	5.71	1.21–27.0
Involvement of intracanalicular segment by OPG	0.031	3.87	1.13–13.2
Involvement of chiasm by OPG	0.041	3.26	1.05–10.1
Post-contrast enhancement of OPG in MRI	0.099	3.71	0.78–17.7
Involvement of optic tracts by OPG	0.115	2.71	0.79–9.34
Endocrinological disorders	0.161	2.96	0.65–13.5
Delayed psychomotor development	0.250	1.93	0.63–5.94
Hypertension	0.260	2.81	0.47–16.9
Attention deficit hyperactivity disorder (ADHD)	0.371	2.22	0.39–12.7
Coincidence of Lisch nodules	0.403	1.61	0.53–4.88
Autism spectrum disorders	0.434	2.68	0.23–31.6
Familial NF1	0.454	1.53	0.50–4.67
Involvement of intraocular segment by OPG	0.823	1.13	0.19–6.50
Allergy	0.923	0.92	0.18–4.66
Involvement of optic nerve by OPG	0.980	0.97	0.11–8.97
Involvement of intraorbital segment by OPG	0.999	ND	ND
Plexiform neurofibroma of any location	0.999	ND	ND
Congenital heart disease	0.999	ND	ND
Hydrocephalus	0.999	ND	ND
Pseudarthrosis	0.999	ND	ND

OR—odds ratio; 95% CI—95% confidence interval; OPG—optic pathway glioma; MRI—magnetic resonance imaging; ND—not done.

**Table 4 cancers-17-01404-t004:** Multivariate analysis for risk factors of oncological therapy.

Parameter	*p*	OR	95% CI
Amblyopia	0.003	9.56	2.19–41.7
Proptosis	0.004	13.9	2.37–82.1
Strabismus	0.106	3.88	0.75–20.1
Bilateral involvement of the visual pathway by OPG	0.122	3.29	0.73–14.9
Epilepsy	0.196	4.59	0.46–6.24
Involvement of intracanalicular segment by OPG	0.222	2.67	0.55–12.9
Thickness of the optic nerve ≥ 8 mm	0.350	2.14	0.43–10.5
Post-contrast enhancement of OPG in MRI	0.66	5.80	0.89–37.9
Involvement of intracranial segment by OPG	0.689	1.59	0.16–15.7
Involvement of chiasm by OPG	0.699	0.71	0.12–4.12

OR—odds ratio; 95% CI—95% confidence interval; OPG—optic pathway glioma; MRI—magnetic resonance imaging.

**Table 5 cancers-17-01404-t005:** Univariate analysis for risk factors of amblyopia.

Parameter	*p*	OR	95%CI
Thickness of the optic nerve ≥ 8 mm	<0.001	5.90	2.21–15.8
Strabismus	<0.001	5.90	2.21–15.8
Delayed psychomotor development	0.002	4.25	1.70–10.7
Involvement of chiasm by OPG	0.006	3.69	1.45–9.36
Bilateral involvement of the visual pathway by OPG	0.007	3.41	1.40–8.32
Involvement of intracranial segment by OPG	0.015	3.23	1.26–8.32
Epilepsy	0.021	12.8	1.48–112
Involvement of intracanalicular segment by OPG	0.048	2.41	1.01–5.76
Familial NF1	0.049	2.42	1.01–5.83
Proptosis	0.056	4.15	0.96–17.9
Endocrinological disorders	0.056	4.15	0.96–17.7
Allergy	0.153	2.38	0.73–7.79
Involvement of optic tracts by OPG	0.290	1.78	0.61–5.17
Post-contrast enhancement of OPG in MRI	0.301	0.62	0.25–1.53
Involvement of intraorbital segment by OPG	0.328	0.65	0.27–1.56
Involvement of intraocular segment by OPG	0.459	0.43	0.05–4.01
Congenital heart disease	0.459	0.43	0.05–4.01
Hypertension	0.461	1.87	0.36–9.82
Hydrocephalus	0.461	1.87	0.36–9.82
Plexiform neurofibroma of any location	0.680	0.77	0.18–2.71
Attention deficit hyperactivity disorder (ADHD)	0.689	1.38	0.29–6.55
Coincidence of Lisch nodules	0.879	0.94	0.39–2.21
Involvement of optic nerve by OPG	0.893	1.13	0.19–6.51
Autism spectrum disorders	0.999	ND	ND
Pseudarthrosis	0.999	ND	ND

OR—odds ratio; 95% CI—95% confidence interval; OPG—optic pathway glioma; NF1—neurofibromatosis type 1; MRI—magnetic resonance imaging; ND—not done.

**Table 6 cancers-17-01404-t006:** Multivariate analysis for risk factors of amblyopia.

Parameter	*p*	OR	95% CI
Strabismus	0.002	8.24	2.22–30.6
Bilateral involvement of the visual pathway by OPG	0.010	4.96	1.46–16.9
Proptosis	0.004	13.96	2.37–82.1
Epilepsy	0.016	22.1	1.77–276
Thickness of the optic nerve ≥ 8 mm	0.033	3.48	1.11–10.9
Involvement of chiasm by OPG	0.284	1.99	0.57–7.03
Familial NF1	0.416	1.60	0.52–4.97
Strabismus	0.350	2.14	0.43–10.5
Involvement of intracanalicular segment by OPG	0.568	1.38	0.46–4.19
Involvement of intracranial segment by OPG	0.585	0.66	0.15–2.90

OR—odds ratio; 95%CI—95% confidence interval; OPG—optic pathway glioma.

**Table 7 cancers-17-01404-t007:** Comparison of our results on NF1-OPG with published data.

NF1-OPG Related Data		Our Results	Comparable Data in Other Authors’ Analyses
1	The typical age of OPG formation is between 2 and 8 years of age.	- Median age at OPG diagnosis was 3.4 years (range: 1.0–24.1 years) - Median age at the time of diagnosis OPG for patients required oncological therapy was 2.9 years (range: 1.0–7.1 years).	Trevisson et al., 2017 [34] Prada et al., 2015 [35]
2	More aggressive OPGs are more common in females than males.	- All OPGs: 57% females (52/92) vs. 43% males (40/92) - Patients with oncological treatment due to progression of OPG: 73% females (11/15) and 27% males (4/15)	Trevisson et al., 2017 [34] Prada et al., 2015 [35] Blazo et al., 2004 [36]
3	The type of mutation in the *Nf1* gene may influence the increased risk of developing OPG, therefore performing genetic testing even in patients meeting clinical criteria is important to better understand the genotype-phenotype correlation.	- Mutations in the *Nf1* gene detected in 21 patients with OPG are presented in Table 1.	Koczkowska et al., 2018 [5] Campen et al., 2018 [9]
4	OPGs may induce pituitary hormonal dysfunction.	- In 7.6% of patients (7/92), hormonal disorders in the form of short stature or precocious puberty were observed - In 1/7 of these patients, OPG was located in the optic chiasm	Azizi et al., 2021 [37]
5	Therapeutic decisions are mainly influenced by the patient’s progressive vision loss.	- Based on multivariate analysis, the statistically significant parameters influencing decisions regarding the implementation of oncological therapy were amblyopia and proptosis	Fisher et al., 2013 [38] Campen et al., 2018 [9]

## Data Availability

The original contributions presented in this study are included in the article. Further inquiries can be directed to the corresponding author.

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
