# Peer review of "Predictors in Optic Pathway Gliomas in Neurofibromatosis Type 1: A Single Center Study"

_cancers, 2025, doi:10.3390/cancers17091404_

Round 1
Reviewer 1 Report
Comments and Suggestions for Authors
Thank you for the opportunity to review this interesting paper with a large number of NF1 patients and OPG.
Please see the following remarks on my behalf:
line 34: strabismus instead of strabismus
line 52: typical of for the disease... please rephrase
lines 69-73: the article would benefit of adding one to two sentences regarding the opthalmological assessment in childhood and difficulties, as well as what it is considered as optimal opthalmological assessment according to age.
line 221: the phrase "The median age of patients at the time of treatment " is not complete. please rephrase
Line 226: please comment on the findings of 16 OPG patients diagnosed before 2 years of age regarding opthalmological, clinical and radiological findings.
Line 229: "Therefore, it seems reasonable to perform the first MRI after the age of 1 year."
The available guideline fron ERN GENTURIS mention MRI after the age of 2 years unless opthalmological assessment is suggestive for OPG or inconclusive/unreliable ophthalmological exam. I would propose rephrasing the above mentioned sentence as there is no other reference supporting MRI at such a young age especially in case of the absence of symptoms.
One of the most important aspects highlighted in relevant articles is the need for multidisciplinary approach in patients with NF1 and OPG. Decision on starting treatment is multifactorial and should be the result of an MDT team. Please comment on the MDT approach.
lines 231-232: there is no consensus on frequency of MRI after diagnosis of an OPG. It depends on age, presence of symptoms and initiation of therapy and among researchers. Follow-up includes definitely include clinical assessment. Please rephrase.
Lines 270-272: are there any relevant publications for NF1 patients treated with Carbo VCR or VBL? please comment
Author Response
COMMENTS 1: Thank you for the opportunity to review this interesting paper with a large number of NF1 patients and OPG. Please see the following remarks on my behalf:
line 34: strabismus instead of striabismus
RESPONSE 1: The correction has been made.
COMMENTS 2: line 52: typical of for the disease... please rephrase
RESPONSE 2: The correction has been made.
COMMENTS 3: lines 69-73: the article would benefit of adding one to two sentences regarding the opthalmological assessment in childhood and difficulties, as well as what it is considered as optimal opthalmological assessment according to age.
RESPONSE 3: Information recommended by the Reviewer has been added to the text.
COMMENTS 4: line 221: the phrase "The median age of patients at the time of treatment " is not complete. please rephrase
RESPONSE 4: There was a mistake (the same text twice). Thank you for your vigilance.
COMMENTS 5: Line 226: please comment on the findings of 16 OPG patients diagnosed before 2 years of age regarding opthalmological, clinical and radiological findings.
RESPONSE 5: Information has been added in section "RESULTS - Characteristics of optic pathway gliomas".
COMMENTS 6: Line 229: "Therefore, it seems reasonable to perform the first MRI after the age of 1 year."
The available guideline fron ERN GENTURIS mention MRI after the age of 2 years unless opthalmological assessment is suggestive for OPG or inconclusive/unreliable ophthalmological exam. I would propose rephrasing the above mentioned sentence as there is no other reference supporting MRI at such a young age especially in case of the absence of symptoms.
RESPONSE 6: Additional commentary has been added to this section of the DISCUSSION.
COMMENTS 7: One of the most important aspects highlighted in relevant articles is the need for multidisciplinary approach in patients with NF1 and OPG. Decision on starting treatment is multifactorial and should be the result of an MDT team. Please comment on the MDT approach.
RESPONSE 7: I agree with Reviewer. Information about MDT treatment decision-making has been added to the INRODUCTION section.
COMMENTS 8: lines 231-232: there is no consensus on frequency of MRI after diagnosis of an OPG. It depends on age, presence of symptoms and initiation of therapy and among researchers. Follow-up includes definitely include clinical assessment. Please rephrase.
RESPONSE 8: Information suggested by the Reviewer has been added.
COMMENTS 9: Lines 270-272: are there any relevant publications for NF1 patients treated with Carbo VCR or VBL? please comment
RESPONSE 9: Information on the effectiveness of VCR+Carbo and VBL in NF1-OPG therapy has been added.
The authors thank the Reviewer for his time and valuable comments.
The relevant modifications are marked in red.
Reviewer 2 Report
Comments and Suggestions for Authors
Predictors in optic pathway gliomas in neurofibromatosis type 1: a single center study
Comments
In this manuscript, the author’s discuss optic pathway gliomas (OPGs), which are common in young patients with neurofibromatosis type 1 (NF1). They analyzed data from 92 patients to assess tumor characteristics, risk factors for amblyopia and oncological treatment, and therapy outcomes. The key findings from the analysis are that OPGs were unilateral in 55.4% of cases and bilateral in 44.6%, with 16.3% of patients requiring oncological treatment. Significant risk factors for treatment included amblyopia and proptosis, while factors contributing to amblyopia included strabismus, epilepsy, and optic nerve thickening. Based on their analysis, the authors propose modifications to current management guidelines, such as reducing MRI frequency in specific cases, avoiding contrast in follow-up MRIs, and adopting a cautious approach to treatment decisions in patients with psychomotor delays or epilepsy. Below are some of the major issue I have which dampens my enthusiasm for this manuscript:
1). The introduction is well-structured providing clear definition of NF-1 and its association with OPGs. It highlights the asymptomatic nature of OPGs and provide clear summary of the diagnostic criteria and chemotherapeutic options along with emerging therapies.
2). However, the introduction lacks some critical analysis of certain points that I feel is important to mention effectively. The paragraph mentioned specific Nf1 mutations associated with OPG risk but does not elaborate on their clinical relevance. The author has explained about NF1 gene mutation he found in 92 patients in table 1, but those are not the same as mentioned here. The introduction also mentioned that OPGs are rarely mortal but does not discuss vision prognosis, quality of life and recurrence rates after treatments. Additionally, more insight into factors affecting tumor progression or predictors of treatment response is needed to better understand the need for the studies and its value for the greater scientific community.
3). The author presents the anatomy of the visual pathway in the introduction without offering any explanation. In sentences 55-56, the author mentions OPGs occurring in various parts of the optic pathway but does not provide details about the pathway itself or the frequency of OPGs in different regions. As a result, this figure seems unnecessary in the manuscript.
4). The material and method section of the manuscript need to be completely revised and re-written so that it is easy for the reader to understand. There is a lack of justification for the study design. The author should provide a brief explanation of why a retrospective design was chosen. Additionally, the author should provide any limitation one should be aware of choosing such study design.
5). The inclusion and exclusion criteria are well established but the total number of patients diagnosed with OPG is not explicitly stated. A flow chart of the sample size and how many were included in the final analysis will be helpful to understand how the sample size was selected.
6). MRI criteria for OPG were briefly mentioned but not detailed. Since it’s a major point in the manuscript a detail analysis of this should be provided by the author.
7). The author presents a comprehensive list of comorbidities; however, the criteria for their assessment remain unclear. Additional details are needed to clarify the selection rationale and the methods used for their evaluation.
8). The statistical approach taken for the study is also not well explained in the material and method section. I want the author to re-write to remove the ambiguity in this section. For example, in the regression model it is not specified with variables are included in the final analysis.
9). The mutations identified in these patients, as presented by the author in Table 1, are not the same as mentioned in the introduction as genetic risk factors. How does the author account for this? Are these mutations which author found in the analysis are common mutations in NF1 gene as a risk factor or are they rear mutations. Or does the author intended to add more NF1 genes mutations as a risk factor for OPGs. A more explanation is needed regarding these otherwise why are these gene mutation are important is not clear to me.
10). In Table 2, the author discusses comorbidities. I would like the author to provide more details about these comorbidities in the introduction or materials and methods section, including relevant literature on their association with NF1 and OPGs. Otherwise, the significance of this information and the new insights it offers to the reader remain unclear.
11). The result section provides detailed breakdown of OPG characteristics like tumor distribution, literality, MRI finding. However, the section present the raw numerical data but does not compare findings with existing literature and lacks discussion on its implication for clinical decision-making.
12). What is the significance of univariate and multivariate analysis. The author should provide a brief interpretation of these statistics for clarity.
13). The discussion jumps between topics for example MRI, atopy, epilepsy and chemotherapy without a clear logical progression. A more structured approach would enhance readability and the point that author tries to make. Some points, such as the role of MRI and the significance of ophthalmologic examinations, are repeated multiple times, making the discussion somewhat redundant.
Comments on the Quality of English LanguageThere are many instances where the sentences are redundant and grammatically incorrect. Kindly streamline the the thoughts and correct the quality of sentences used.
Author Response
COMMENTS 1: In this manuscript, the author’s discuss optic pathway gliomas (OPGs), which are common in young patients with neurofibromatosis type 1 (NF1). They analyzed data from 92 patients to assess tumor characteristics, risk factors for amblyopia and oncological treatment, and therapy outcomes. The key findings from the analysis are that OPGs were unilateral in 55.4% of cases and bilateral in 44.6%, with 16.3% of patients requiring oncological treatment. Significant risk factors for treatment included amblyopia and proptosis, while factors contributing to amblyopia included strabismus, epilepsy, and optic nerve thickening. Based on their analysis, the authors propose modifications to current management guidelines, such as reducing MRI frequency in specific cases, avoiding contrast in follow-up MRIs, and adopting a cautious approach to treatment decisions in patients with psychomotor delays or epilepsy. Below are some of the major issue I have which dampens my enthusiasm for this manuscript:
The introduction is well-structured providing clear definition of NF-1 and its association with OPGs. It highlights the asymptomatic nature of OPGs and provide clear summary of the diagnostic criteria and chemotherapeutic options along with emerging therapies. However, the introduction lacks some critical analysis of certain points that I feel is important to mention effectively. The paragraph mentioned specific Nf1 mutations associated with OPG risk but does not elaborate on their clinical relevance.
RESPONSE 1: In the cited publication, mutations with a statistically proven effect on the formation of NF1-OPG were only mentioned. Unfortunately, there is no information about the aggressiveness of these tumors and their clinical significance. It was only shown that the presence of these mutations correlates with an increased risk of OPG.
COMMENTS 2: The author has explained about NF1 gene mutation he found in 92 patients in table 1, but those are not the same as mentioned here.
RESPONSE 2: We have identified the genotype of 21/92 patients presented in our study. We did not find the occurrence of these mutations in other analyses related to NF1-OPG. However, due to the need to learn about genotype-phenotype correlations, we have thoroughly analyzed the course of OPG in patients with identified mutations, in order to possibly use these data in further analyses focused on genotype-phenotype correlations of patients with NF1.
COMMENTS 3: The introduction also mentioned that OPGs are rarely mortal but does not discuss vision prognosis, quality of life and recurrence rates after treatments. Additionally, more insight into factors affecting tumor progression or predictors of treatment response is needed to better understand the need for the studies and its value for the greater scientific community.
RESPONSE 3: Information suggested by the Reviewer has been added.
COMMENTS 4: The author presents the anatomy of the visual pathway in the introduction without offering any explanation. In sentences 55-56, the author mentions OPGs occurring in various parts of the optic pathway but does not provide details about the pathway itself or the frequency of OPGs in different regions. As a result, this figure seems unnecessary in the manuscript.
RESPONSE 4: The authors remind readers of the structure of the optic pathway so that they can better understand the results presented in the following section. In the section "RESULTS - Characteristics of optic pathway gliomas" the authors precisely present the involvement of all segments of the optic pathway in the analyzed study group.
COMMENTS 5: The material and method section of the manuscript need to be completely revised and re-written so that it is easy for the reader to understand. There is a lack of justification for the study design. The author should provide a brief explanation of why a retrospective design was chosen. Additionally, the author should provide any limitation one should be aware of choosing such study design.
RESPONSE 5: In the "MATERIALS AND METHODS - Patients" section the study design is described. In the DISCUSSION section the limitations of the conducted analysis are discussed.
COMMENTS 6: The inclusion and exclusion criteria are well established but the total number of patients diagnosed with OPG is not explicitly stated. A flow chart of the sample size and how many were included in the final analysis will be helpful to understand how the sample size was selected.
RESPONSE 6: A figure explaining the selection of patients for analysis has been added.
COMMENTS 7: MRI criteria for OPG were briefly mentioned but not detailed. Since it’s a major point in the manuscript a detail analysis of this should be provided by the author.
RESPONSE 7: Following the valuable suggestion of the Reviewer, the criteria for diagnosing OPG in MRI have been added.
COMMENTS 8: The author presents a comprehensive list of comorbidities; however, the criteria for their assessment remain unclear. Additional details are needed to clarify the selection rationale and the methods used for their evaluation.
RESPONSE 8: We analyzed all available clinical data of our patients, as described in the methodology:
"We analyzed patient demographics, imaging and ophthalmological examination results, and their impact on further therapeutic decisions and the effectiveness of used therapy. Additionally, all available clinical and genetic data were assessed. "
COMMENTS 9: The statistical approach taken for the study is also not well explained in the material and method section. I want the author to re-write to remove the ambiguity in this section. For example, in the regression model it is not specified with variables are included in the final analysis.
RESPONSE 9: The „Statistical analysis” section was corrected and supplemented. Respective chenges are marked in revised version of the manuscript.
COMMENTS 10: The mutations identified in these patients, as presented by the author in Table 1, are not the same as mentioned in the introduction as genetic risk factors. How does the author account for this? Are these mutations which author found in the analysis are common mutations in NF1 gene as a risk factor or are they rear mutations. Or does the author intended to add more NF1 genes mutations as a risk factor for OPGs. A more explanation is needed regarding these otherwise why are these gene mutation are important is not clear to me.
RESPONSE 10: We have identified the genotype of 21/92 patients presented in our study. We did not find the occurrence of these mutations in other analyses related to NF1-OPG. However, due to the need to learn about genotype-phenotype correlations, we have thoroughly analyzed the course of OPG in patients with identified mutations, in order to possibly use these data in further analyses focused on genotype-phenotype correlations of patients with NF1.
COMMENTS 11: In Table 2, the author discusses comorbidities. I would like the author to provide more details about these comorbidities in the introduction or materials and methods section, including relevant literature on their association with NF1 and OPGs. Otherwise, the significance of this information and the new insights it offers to the reader remain unclear.
RESPONSE 11: Reference to the comorbidity of OPG in NF1 patients has been added in the "ITRODUCTION" section. Additionally, selected comorbidities are discussed in the DISCUSSION.
COMMENTS 12: The result section provides detailed breakdown of OPG characteristics like tumor distribution, literality, MRI finding. However, the section present the raw numerical data but does not compare findings with existing literature and lacks discussion on its implication for clinical decision-making.
RESPONSE 12: One of the main goals of our work was to analyze the influence of tumor anatomy on their aggressiveness, risk of vision loss and other symptoms such as proptosis, strabismus or hormonal disorders. Additionally, due to the high need to learn about genotype-phenotype correlations, we provided the frequency of all diseases coexisting with OPG in the study group and listed patients who underwent genetic analysis. This information was added to the "MATERIALS AND METHODS" section.
COMMENTS 13: What is the significance of univariate and multivariate analysis. The author should provide a brief interpretation of these statistics for clarity.
RESPONSE 13: For interpretation of results of multivariate analysis an additional section was created in „Discussion” section.
COMMENTS 14: The discussion jumps between topics for example MRI, atopy, epilepsy and chemotherapy without a clear logical progression. A more structured approach would enhance readability and the point that author tries to make. Some points, such as the role of MRI and the significance of ophthalmologic examinations, are repeated multiple times, making the discussion somewhat redundant.
RESPONSE 14: The arrangement of "Discussion" in the order: epidemiology, diagnostics, clinical symptoms, treatment has been modified. Duplicate items have been removed.
COMMENTS 15: Comments on the Quality of English Language
There are many instances where the sentences are redundant and grammatically incorrect. Kindly streamline the the thoughts and correct the quality of sentences used.
RESPONSE 15: The quality of the text and language has been improved.
The authors thank the Reviewer for his time and valuable comments.
The relevant modifications are marked in green.
Reviewer 3 Report
Comments and Suggestions for Authors
Marjanska et al. should be commended for a very comprehensive paper retrospectively looking at demographics, clinical and genetic data, imaging, and ophthalmological parameters impacting therapeutic decisions and the effectiveness of therapy in NF1-related optic pathway gliomas. This was a very interesting paper, with some nice suggestions to refine clinical practice. Please consider adding the following data to the paper.
- What are the mortality and causes of death in OPG patients?
- How long did patients receive chemotherapy?
- How old were patients whose tumors went into senescence?
Author Response
Marjanska et al. should be commended for a very comprehensive paper retrospectively looking at demographics, clinical and genetic data, imaging, and ophthalmological parameters impacting therapeutic decisions and the effectiveness of therapy in NF1-related optic pathway gliomas. This was a very interesting paper, with some nice suggestions to refine clinical practice. Please consider adding the following data to the paper.
COMMENTS 1: What are the mortality and causes of death in OPG patients?
RESPONSE 1: Information about mortality and causes of death in OPG patients has been added in the INTRODUCTION section.
COMMENTS 2: How long did patients receive chemotherapy?
RESPONSE 2: As suggested by the Reviewer, information on duration of therapy has been added in the "RESULTS - Oncological therapy" section.
COMMENTS 3: How old were patients whose tumors went into senescence?
RESPONSE 3: The data, as suggested by the Reviewer, have been added.
We would like to thank the Reviewer for his time and very nice opinion.
The relevant modifications are marked in blue.
Round 2
Reviewer 1 Report
Comments and Suggestions for Authors
I would like to thank the authors for taking into consideration my suggestions/comments.
A minor comment:
line73: MRI show... please rephrase
The manuscript's content has undergone significant improvement. t opportunity yo
Author Response
COMMENTS: line73: MRI show... please rephrase
RESPONSE: The sentence has been changed according to the Reviewer's suggestion.
Once again, I would like to thank the Reviewer for his time.
Reviewer 2 Report
Comments and Suggestions for Authors
My suggestion to revise the Materials and Methods section stems from the fact that it remains difficult to follow and lacks sufficient detail to fully support and contextualize the results presented. Despite the revisions made so far, the section still appears underdeveloped, limiting our ability to interpret the findings effectively. Furthermore, I remain unsatisfied with the response regarding the gene mutations listed in Table 1, partly because the methodological details relevant to this aspect are not clearly articulated. As you have mentioned that you wanted to determine the genotype-phenotype correlation, hence the clarity regarding gene mutation become imperative.
Author Response
COMMENTS: My suggestion to revise the Materials and Methods section stems from the fact that it remains difficult to follow and lacks sufficient detail to fully support and contextualize the results presented. Despite the revisions made so far, the section still appears underdeveloped, limiting our ability to interpret the findings effectively. Furthermore, I remain unsatisfied with the response regarding the gene mutations listed in Table 1, partly because the methodological details relevant to this aspect are not clearly articulated. As you have mentioned that you wanted to determine the genotype-phenotype correlation, hence the clarity regarding gene mutation become imperative.
RESPONSE:
- We reorganized the "MATERIALS AND METHODS" section.
- We added a new section "Analyzed factors" in the methodology.
- We added primary and secondary endpoints of the study.
- Due to the lack of possibility to find genotype-phenotype correlations, we modified the objective of the study. We presented mutations of our patients with OPG as unique data.